

# Application of deep autoencoder as an one-class classifier for unsupervised network intrusion detection: a comparative evaluation

Thavavel Vaiyapuri and Adel Binbusayyis

College of Computer Engineering and Sciences, Prince Sattam bin Abdulaziz University, AlKharj, Saudi Arabia

## ABSTRACT

The ever-increasing use of internet has opened a new avenue for cybercriminals, alarming the online businesses and organization to stay ahead of evolving thread landscape. To this end, intrusion detection system (IDS) is deemed as a promising defensive mechanism to ensure network security. Recently, deep learning has gained ground in the field of intrusion detection but majority of progress has been witnessed on supervised learning which requires adequate labeled data for training. In real practice, labeling the high volume of network traffic is laborious and error prone. Intuitively, unsupervised deep learning approaches has received gaining momentum. Specifically, the advances in deep learning has endowed autoencoder (AE) with greater ability for data reconstruction to learn the robust feature representation from massive amount of data. Notwithstanding, there is no study that evaluates the potential of different AE variants as one-class classifier for intrusion detection. This study fills this gap of knowledge presenting a comparative evaluation of different AE variants for one-class unsupervised intrusion detection. For this research, the evaluation includes five different variants of AE such as Stacked AE, Sparse AE, Denoising AE, Contractive AE and Convolutional AE. Further, the study intents to conduct a fair comparison establishing a unified network configuration and training scheme for all variants over the common benchmark datasets, NSL-KDD and UNSW-NB15. The comparative evaluation study provides a valuable insight on how different AE variants can be used as one-class classifier to build an effective unsupervised IDS. The outcome of this study will be of great interest to the network security community as it provides a promising path for building effective IDS based on deep learning approaches alleviating the need for adequate and diverse intrusion network traffic behavior.

Corresponding author
Thavavel Vaiyapuri,
t.thangam@psau.edu.sa

## INTRODUCTION

The advances in networking technologies have fueled the significance of the Internet in various domains of human society. While the Internet is finding a global reach, cybercriminals are becoming even more proficient in looting the benefits of Internet

openness to advance their attacks at an alarming pace (*Binbusayyis & Vaiyapuri, 2019*). The increase in scope and severity of cyberattacks than ever before is alarming the online businesses and organizations to stay ahead of cybercriminals. Under this background, the intrusion detection system (IDS) is deemed as one of the most promising defensive mechanism of cybersecurity and has attracted lot of research attention recently in the field of network security (*Binbusayyis & Vaiyapuri, 2020*). Although IDS have evolved to a highly sophisticated level, their detection capability is confronted with massive increase in network traffic volume and complex network structure resulting from today's extreme use of Internet. Thenceforth, there is a surge of interest in devolving new effective approach that learn the most robust feature representation from massive network data to step up the detection accuracy of an IDS.

The recent technical breakthroughs in deep learning ability (*Aldweesh, Derhab & Emam, 2020*) for feature representation with large volume of data have sparked a revolution in designing effective IDS to achieve new performance level and safeguard the computer networks from thwart cyberattacks arising from ever changing threat landscape. While these methods have gained ground in the field of intrusion detection, notwithstanding, the majority of progress has been witnessed on supervised tasks which requires adequate and diverse labeled data for training. But in real network environment, the process of labeling large volume of network traffic data is laborious and error-prone. In light of these limitations, the development of IDS based unsupervised deep learning approaches is gaining more attention with vital practical importance (*Choi et al., 2019*). These approaches do not demand labeled data for training and have potential to detect intrusion activities in the network traffic without any prior knowledge about the intrusion behavior.

Amongst various unsupervised deep learning approaches, autoencoder (AE) has been extensively explored in the field of intrusion detection on the grounds that it enables data-driven approach for robust feature representation from massive amount of data and exhibits strong ability for data reconstruction. For example, similar to our study, *Choi et al. (2019)* investigated the performance of different AE variant within unsupervised learning framework. But unlike the work discussed here, they train the AE network in a supervised manner using both normal and abnormal samples to set the reconstruction error (RE) value as a heuristic threshold to improve the detection accuracy of IDS. Likewise, another study by *Naseer et al. (2018)* examined the suitability of deep learning approaches for intrusion detection. Toward this, they developed different deep learning model-based IDS and utilized AE for capturing robust feature representation that can enhance the discrimination ability of IDS classifier. Also *Aygun & Yavuz (2017)* attempted to enhance the AE discriminative ability utilizing stochastically determined threshold for RE to reach an improved accuracy compared to deterministic AE variants on NSL-KDD intrusion datasets. In the same manner, *Ieracitano et al. (2020)* presented statistical analysis to extract more optimized and correlated features to improve the accuracy of AE. Also, *Shone et al. (2018)* recently introduced non-symmetric variant of deep AE for unsupervised learning and achieved promising results.

The above examined literature ascertains that considerable progress has been made to utilize the potential of different variants of AE to improve the detection accuracy of IDS. Nonetheless, there is no concrete study that examines and compares the potential of these variants of AE as one-class classifier for unsupervised IDS. To fill this gap of knowledge, this study for the first time accounts to provide an experimental comparison on detection performance between different AE variants on the development of unsupervised IDS. The variants selected for comparison includes Stacked AE (SAE), Sparse AE (SSAE), Denoising AE (DAE), Contractive AE (ContAE) and Convolutional AE (CAE). The reason for choosing these variants is that they are competitive based on the performance reported in their corresponding literature. Added on, currently they are most commonly used in practice (*Bayram, Duman & Ince, 2020*; *Abirami & Chitra, 2020*). The variants are evaluated on different benchmark datasets, NSL-KDD and UNSW-NV15 considering comprehensive evaluation metrics. A unified network configuration is adopted for all variants to ensure uniform model complexity and make a meaning comparison. The extensive evaluation results demonstrate the respective benefits and generalization ability of different AE variants with regard to attacks of all kinds. This evaluation is expected to provide a valuable insight on how the potential of different AE variants can be used to develop an effective unsupervised IDS.

## AUTOENCODER FOR ONE-CLASS CLASSIFICATION

An AE is neural network that learns the intrinsic network traffic features reconstructing the original network traffic at its output layer (*Rumelhart, Hinton & Williams, 1986*). As shown in Fig. 1, the architecture of an AE consists of two key networks, encoder and decoder connected in serial. As represented by Eq. (1), the encoder network generates the feature representation by mapping the given input network traffic to hidden layer using an activation function $f$ parameterized by $W$ and $b$.

$$H = f(WX + b) \tag{1}$$

Similarly, the decoder network reconstructs the original input network traffic from the generated feature representation using the activation function $g$ parameterized by $W'$ and $b'$ as given below

$$Z = g(W'H + b') \tag{2}$$

The AE is trained jointly with given training samples to learn the parameter set $\theta = \{W, W', b, b'\}$ of two networks, encoder and decoder minimizing the RE. Concretely, the cost function of the AE is defined as follows,

$$J_{AE}(\theta) = \min_{\theta} \frac{1}{2N} \|X - Z\|^2 + \frac{\lambda}{2} (\|W\|^2 + \|W'\|^2) \tag{3}$$

Here the first term measures the RE between original input and reconstructed output data over $N$ input samples, the second term called regularization term is employed to restrain the magnitude of the weights and help thwart the network from overfitting, and $\lambda$ is a weight decay parameter that controls the proportion of regularization and RE.

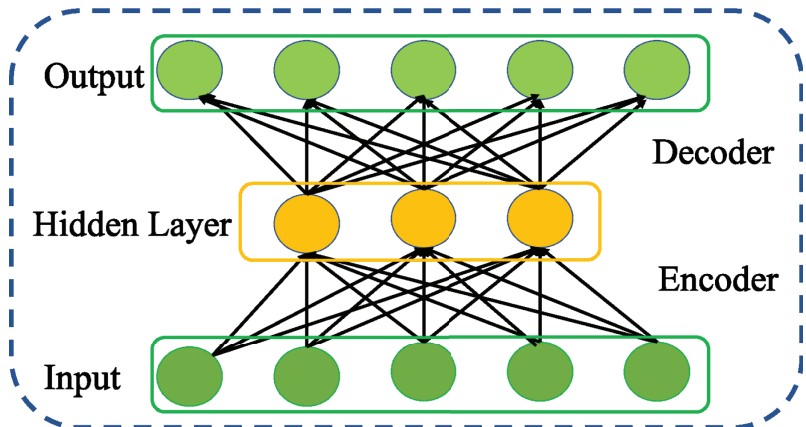

**Figure 1  Structure of general AE network.**

In 2014, *Sakurada & Yairi (2014)* devised a holistic approach for anomaly detection utilizing the reconstruction ability of the AE. The basic intuition behind this approach is that AE trained only with normal samples will fail to reconstruct abnormal or malicious samples that it has not confronted earlier and will display high RE. Evidently, the RE of AE can be used as indicator for anomaly detection and can be employed as an alternative for one-class classification. Since then, AE is being widely applied for anomaly detection in various domain and intrusion detection is not an exception.

For instance, the acquisition of network traffic with various anomalous behavior is practically impossible in most real networking environments. Under such circumstances, the application of AE for one-class classification in the field of intrusion detection will play a significant role in successfully modeling the normal network traffic behavior. In this perspective, we aim to examine significance of AE for intrusion detection. Towards this end, Algorithm 1 illustrates how AE can be utilized for one-class classification to detect any kind of attacks and its working principles is depicted in Fig. 2. The average (Avg) shown in Fig. 2 is determined during the training process of AE by computing the average of RE over all training samples and is utilized as a threshold ($\alpha$) to detect the intrusion from normal traffic data.

# VARIANTS OF AUTOENCODER

## Stacked autoencoder

A SAE is a common variant of AE. It is called as Deep AE as it is constructed stacking multiple AEs successively such that the output of first AE is fed as input to next AE and so on as shown in Fig. 3. Since in practice, it is tedious task to train all AEs simultaneously, SAE adopts greedy layer-wise training in forward order where each AE learn their parameters reconstructing the output of the previous AE. The output of the $k^{\text{th}}$ AE is computed as follows setting $H^0 = X$.

$$H^k = f(W^k H^{k-1} + b^k) \tag{4}$$

**Algorithm 1  AE as one-class classifier**

**Input:** X-Training and Testing Set

**Output:** Classification Results

**Initialize:** all AE network parameters

**Procedure:**

  **Phase-I: Training**

  **for each** training epoch **do**

    **for each** mini batch **do**

      1. $H \leftarrow$ Feature representation using Eq. (1)

      2. $Z \leftarrow$ Reconstructed Input using Eq. (2)

      3. Compute the gradient to minimize Cost function in Eq. (3) using Adam

      4. Update AE network parameters

    **end for**

  **end for**

  $\alpha \leftarrow$ average Reconstruction Error on Training data set

**Phase-II: Testing**

**for each** sample in Testing dataset **do**

    1. *Error* $\leftarrow$ Compute reconstruction Loss

    2. *if Error* $> \alpha$ **then** Sample is attack *else* Sample is normal traffic

**end for**

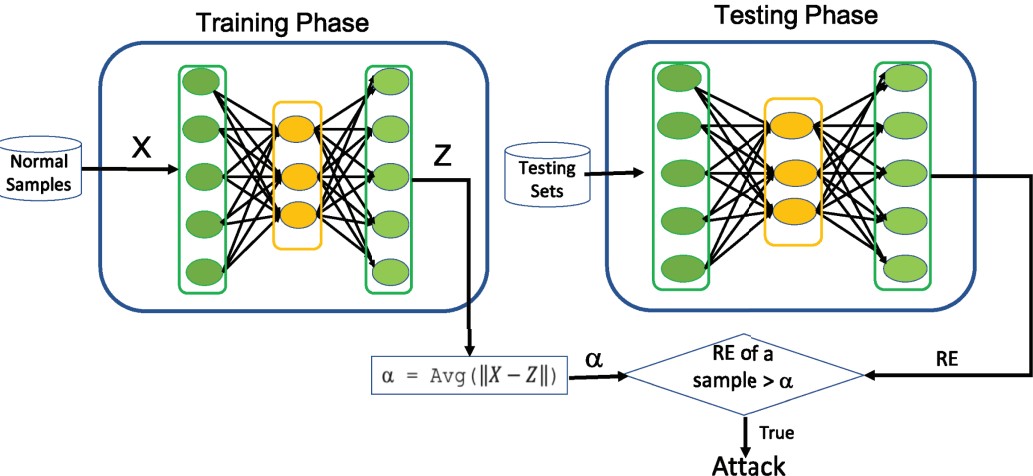

**Figure 2  Illustration of using AE as one-class Classifier.**

    Thus, the layer-wise training enables SAE to learn more abstract and essential information gradually (*Bengio et al., 2007*). After training is completed, the encoding layer in all AEs are concatenated and followed by decoding layer of all AEs. Thus, SAE consists

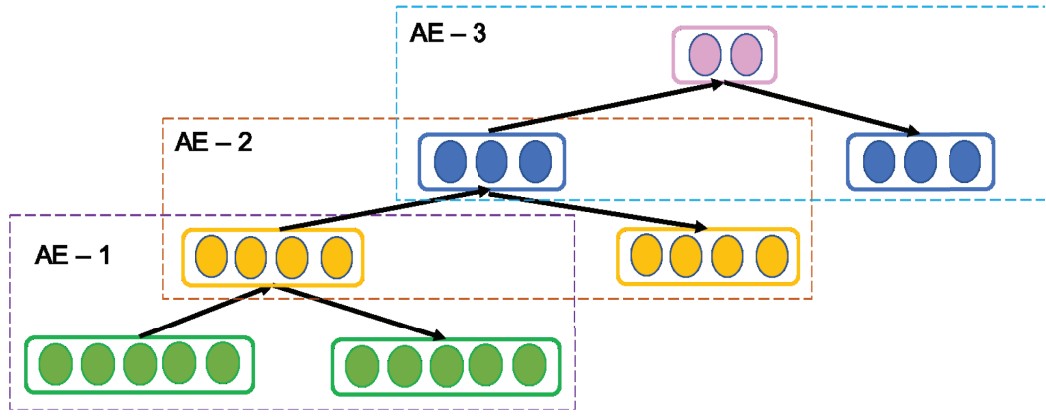

**Figure 3 Structure of general Stacked AE (SAE) network.**

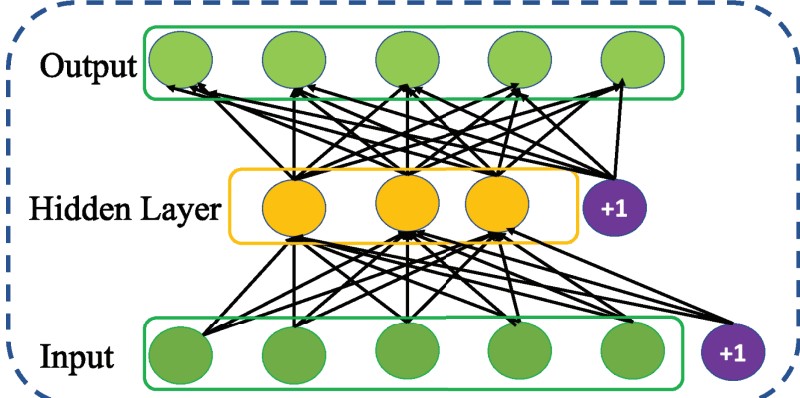

**Figure 4 Structure of general Sparse AE (SSAE) network.**

of multiple hidden layers to drive deep representative features of the inputs that supports in enriching its reconstruction ability.

## Sparse autoencoder

Sparse autoencoder is a variation of basic AE proposed by *Ranzato et al. (2007)*. It aims to learn sparse features of the data introducing a sparse constraint to the core idea of basic AE. The act of imposing sparse constraint on hidden layer as shown in Fig. 4 restricts the undesired activation and enables to maintain low average activation of hidden units. This encourages SSAE for improved learning of sparse features at its hidden representation. Taking the sparse constraint into account, the cost function of SSAE is revised as follows,

$$J_{SSAE}(\theta) = J_{AE}(\theta) + \alpha \sum_{j=1}^{s} KL(\rho \| \hat{\rho}) \tag{5}$$

Thus, the Kullback–Leibler (KL) divergence between desired $\hat{\rho}$ and actual $\rho$ distribution is used to add the sparse constraint as a regularizer to the cost function of AE and the

**Peer**J Computer Science

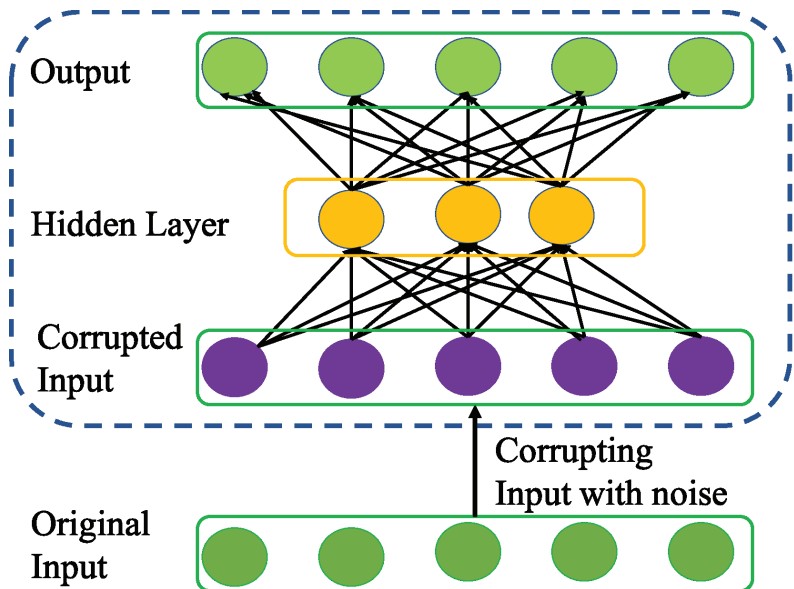

**Figure 5  Structure of general Denoising AE (DAE) network.**

parameter α is used control the relative importance of sparsity penalty term in the cost function.

## Denoising autoencoder

*Vincent et al. (2008)* introduced a variation of AE called DAE to enhance the robustness of basic AE for learning more generalizable features from noisy input data and help the network prevent overfitting. To achieve this, DAE as a first step utilizes stochastic mapping to corrupt the original input $x$ as follows $\hat{x} = p(\hat{x}|x)$.

Then, DAE inherits the core idea of AE to encode the corrupted input $\hat{x}$ to a hidden representation as follows.

$$H = f(W\hat{X} + b) \tag{6}$$

Unlike basic AE, the encoding process in DAE is very effective in capturing the more representative features that can nullify the effect of corruption for better data reconstruction as shown in Fig. 5. Apparently, the advantage of DAE is two-fold: First, it can capture and eliminate the statistical dependencies among the inputs for better feature representation learning. Second, it displays stable performance especially when the input is corrupted with noise.

## Contractive autoencoder

*Rifai et al. (2011)* followed up DAE and introduced a variation of AE called ContAE as an effective solution for improving the learning robustness of basic AE against the influence of noise perturbation on input data. To this end, a contractive penalty is adopted with Frobenius norm of Jacobian square summation of all partial derivatives of the hidden representation in relation to the input data which is represented as follows,

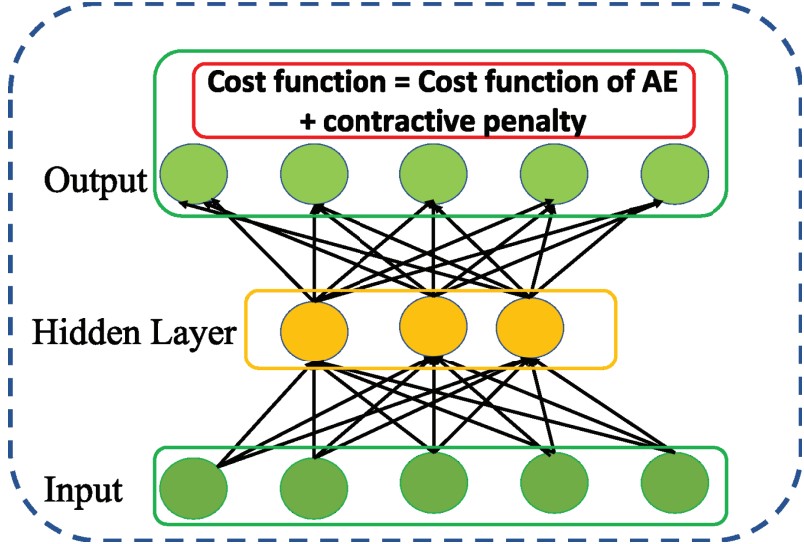

**Figure 6 Structure of general Contractive AE (ContAE) network.**

$$\left\|J_f(x)\right\|_F^2 = \sum_{ij} \left(\frac{\partial h_j(x)}{\partial x_i}\right)^2 \tag{7}$$

The above penalizing term tends to imply that the learnt features are locally invariant and insensitive to small changes in input data. Intuitively, the cost function of ContAE with penalizing term is written as follows,

$$J_{ContAE}(\theta) = J_{AE}(\theta) + \left\|J_f(x)\right\|_F^2 \tag{8}$$

Although both DAE and ContAE share same motivation of improve the robustness of basic AE, the idea adopted are distinct. For instance, DAE achieves stochastically by corrupting the input with random noise. Whereas, ContAE achieves analytically by balancing the RE with contractive penalty term as shown in Fig. 6.

## Convolutional autoencoder

*Masci et al. (2011)* leveraging the benefits of CNN and AE proposed CAE to achieve strong feature representation. Compared to other variants of AE, CAE accomplishes strong feature representation considering the relationships among the features that are more appropriate to eliminate irrelevant and redundant features. Further, CAE enables weight sharing among the inputs and ensures to preserve the spatial locality of the features. Doing so, the number of parameters to be trained is reduced. This in turn reduces the memory requirement and computational efficiency of CAE. Thus, CAE is regarded as special type of AE with convolutional layer as shown in Fig. 7 rather than fully connected layer for encoding process and deconvolutional layer for decoding process.

Taking the inspiration from *Chen, Yu & Wang (2020)*, this study adopts 1D CAE with a hypothesis that the application of 1D CAE will enable to achieve further higher efficiency with the sequential form of network traffic data compared to 2D CAE.

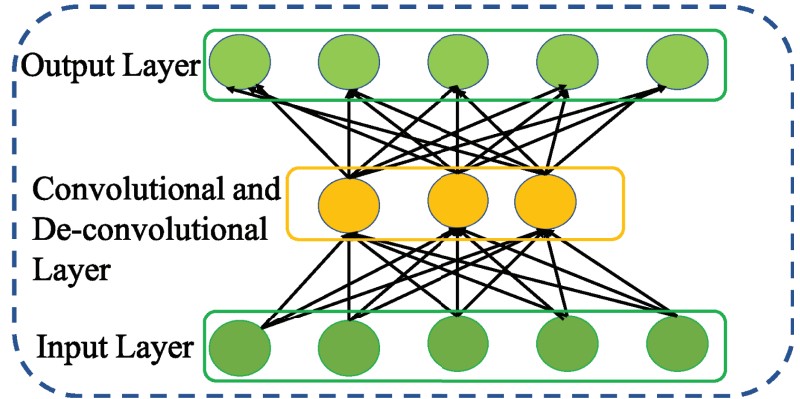

**Figure 7 Structure of general Convolutional AE (CAE).**

Accordingly, the encoding process for convolutional layer with feature filters, maps 1D input data $X$ to produce hidden representation with $k^{th}$ feature map is given as,

$$H_k = f(X * W_k + b_k) \tag{9}$$

Here $*$ denotes 1D convolution operation over the input vector $X$. Similarly, the decoding process with deconvolution operation is represented as follows

$$Z = f(H_k * W_k^T + b_k') \tag{10}$$

## EXPERIMENTAL SETUP

This section first introduces the common network architecture utilized to compare all AEs. Subsequently, it describes the experimental datasets and the framework adopted for comparative evaluation. Finally, the implementation is presented.

### Designed AE network structure

To conduct fair and valid comparison, a common network architecture that displayed reasonable results with all the chosen AE variants over all datasets is determined conducting a range of preliminary experiments. In this determination process, RE was used as an indicator to obtain the best performing network architecture, which is illustrated in Table 1. This architecture consists of two hidden layers of dimension 32 and 24 respectively in the encoder side and a bottleneck layer of dimension 16. Besides, Tanh is used as activation function in all layers.

From the common network architecture, SSAE is developed by attaching the sparse regularization on hidden layers to ensure acquisition of unique statistical features from the given input dataset. The DAE is implemented by corrupting the preprocessed input data with gaussian noise of level 10%. Since, the performance of DAE decreases as the noise level increases, especially when σ > 0.5. This is mainly because the higher noise level will cause the loss of useful information, resulting in difficult reconstruction and poor classification performances.

**Table 1 Structural parameters of Deep AE.**

| Parameter | Value |
| --- | --- |
| Input Layer Dimension | (41,1) |
| Number of Hidden Layer in Encoder | 3 |
| Number of nodes in 1st Hidden Layer | 32 |
| Number of nodes in 2st Hidden Layer | 24 |
| Number of nodes in bottleneck Layer | 16 |
| Activation function | Tanh |
| Sparsity Penalty term in SSAE in hidden layer | 1e−5 |
| Sparsity Penalty term in SSAE in bottleneck layer | 1e−4 |
| Contractive Penalty term in ContAE | 1e−5 |
| Gaussian noise corruption factor in DAE | 0.1 |

**Table 2 Structural parameters of CAE.**

| Layer type | Input size | Output size |
| --- | --- | --- |
| Input | (41,1) | (41,1) |
| Conv1D | (41,1) | (41,8) |
| BatchNormalization | (41,8) | (41,8) |
| Maxpooling1D | (41,8) | (20,8) |
| Conv1D | (20,8) | (20,8) |
| BatchNormalization | (20,8) | (20,8) |
| Upsampling1D | (20,8) | (40,8) |
| Conv1D | (40,8) | (40,8) |
| BatchNormalization | (40,8) | (40,8) |
| Conv1DTranspose | (40,8) | (41,1) |

The ContAE is developed adding the contractive penalty term to the cost function of SAE. To develop CAE, the hidden layers and bottleneck layers in the common network architecture are replaced with convolutional, batch normalization as illustrated in Table 2. Further, the number and size of kernels is kept same for all convolution layers and is set to 8 and $1 \times 3$ respectively. This pyramid architecture not only reduces the number of trainable parameters but also enables to learn the most essential features from input network traffic by eliminating the redundant and irrelevant features. Besides, max pooling layer with pool size of 2 is employed to extract the most essential features from the input data and upsampling layer of pool size 2 is used to reconstruct the original input from the extracted essential features.

## Datasets

A number of datasets are available publicly for IDS research evaluation. Nonetheless, these datasets suffer from absences of traffic diversity and lack of sufficient number of sophisticated attack styles. Therefore, in order to conduct a fair and effective evaluation of the proposed model, an old benchmark NSL-KDD dataset and a new contemporary

**Table 3 Data distribution in NSL-KDD.**

| Class | Training set | Testing set |
|---|---|---|
| Normal | 67,343 | 9,710 |
| Attack | 58,630 | 12,833 |
| Total | 125,973 | 22,543 |

**Table 4 Data distribution in UNSW-NB15.**

| Class | Training set | Testing set |
|---|---|---|
| Normal | 56,000 | 37,000 |
| Attack | 119,341 | 45,332 |
| Total | 175,341 | 82,332 |

UNSW-NB15 dataset are considered in this work. A brief description of these two intrusion datasets is given below.

### NSL-KDD dataset

The NSL-KDD dataset is an improved version of KDD'99 dataset, presented by *Tavallaee et al. (2009)* resolving the redundancy in KDD'99 dataset. This dataset contains an optimal ratio of 125,973 training samples to 22,543 testing samples. Thus NSL-KDD is regarded as one of the most valuable benchmark resource in the field cybersecurity research for IDS evaluation. Each sample in NSL-KDD contains 41 features and 1 class label to characterize whether the network traffic is normal or belongs to attack category. The distribution of normal traffic samples in the training and testing sets with regard to attacks are given in Table 3.

### UNSW-NB15 dataset

The UNSW-NB15 is a modernized dataset recently developed by ACCS with hybrid of real normal and synthesized contemporary attack behavior from network traffic flow (*Moustafa & Slay, 2015*). This dataset includes 9 families of attacks namely DoS, Analysis, Generic, Fuzzers, Backdoors, Exploits, Shellcode, Reconnaissance, and Worms. The dataset consists of 175,341 training samples and 82,332 testing samples, each characterized with 42 features and a class label to discriminate the network traffic as normal or malicious activities. The distribution of samples against normal and attack class is shown in Table 4.

## Designed experimental framework for comparison of AE variants

The experimental framework designed for this comparative study is shown in Fig. 8. It involves three steps: First, the raw network traffic data are prepared for subsequent processing. This includes two main preprocessing operations, symbolic value encoding and normalization. Next, Only the normal network traffic samples from the preprocessed training dataset is employed to train all AE variants. Further, as an evaluation protocol, 5-fold cross-validation strategy recommended in literature is adopted on all AE variants

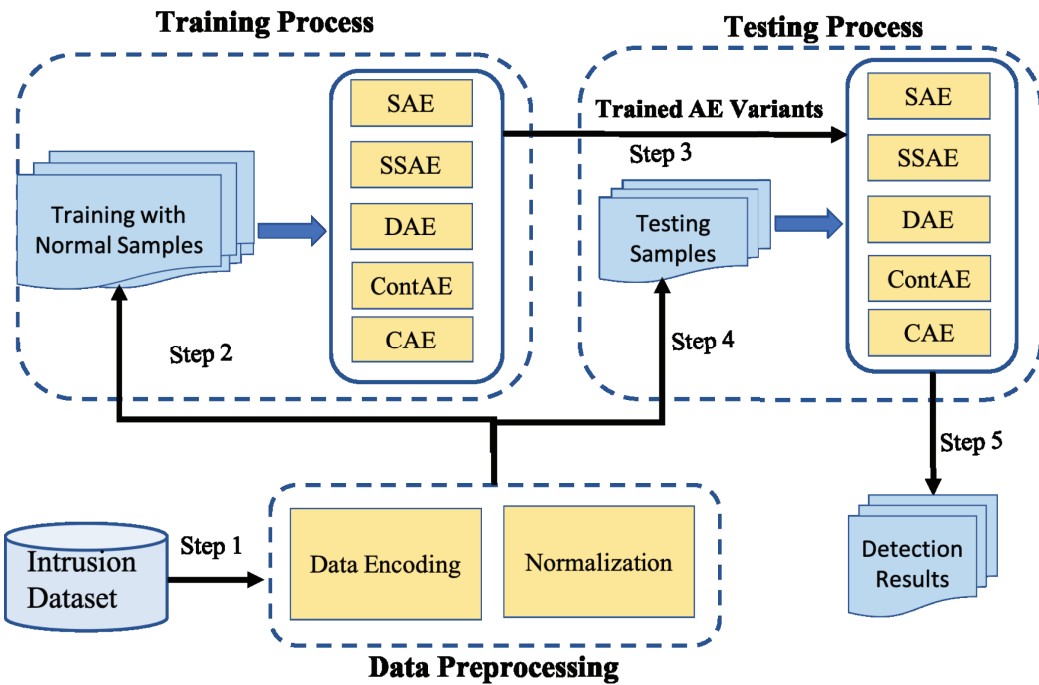

**Figure 8  Illustration of experimental framework.**

to prevent overfitting and manage accurately the trade-off between bias and variance. After training process is completed, the preprocessed testing dataset is employed into the trained AE variants and their performance is compared against the comprehensive set of metrics.

### Data preprocessing

Data preprocessing is an essential task in ensuring the quality of the input for the subsequent model training and testing process. Thereof, both the training and testing dataset are subjected to the following two preprocessing operations in sequence to boost the performance of all chosen AEs for intrusion detection.

a) *Data Encoding:* This operation encodes all non-numeric or nominal features in the given dataset to numeric values. Here, a nominal feature with $C$ different values is encoded with numeric values ranging from 0 to $C-1$.

b) *Normalization:* Generally, the machine learning algorithms are biased by input features with large numeric value. To combat this effect, min-max normalization is applied to adjust the value range of all input features within the range [0,1].

### Training process

After data preprocessing task, all the chosen AE variants are trained to learn their network model parameters optimizing their respective cost function using Adam as stochastic gradient optimization method, because of its adaptability and computational efficiency (*Da, 2014*). Further to ensure a fair comparison, all the chosen AE variants are

**Table 5 Training parameters for AE.**

| Parameter | Value |
|---|---|
| Optimizer | Adam |
| Batch Size | 128 |
| Number of epochs | 20 |
| Validation | 5-fold cross validation |

trained with batch size of 128 for 20 epochs using a learning rate of 0.001. Besides, the trainable parameters of all AE variants are initialized applying Xavier algorithm to keep the backpropagated gradient and activation value within a reasonable range (*Glorot & Bengio, 2010*). Table 5 presents the parameters used for training all the chosen AE variants.

### Evaluation metrics

Once all the chosen AE variants are trained successfully, they are evaluated on pre-proceed testing dataset for intrusion detection. As discussed in "Autoencoder For One-Class Classification", the trained AE detects the intrusion network traffic samples on the basis that the RE will be larger for intrusion traffic than that for normal one. To implement this, a threshold value is defined for each AE based on their respective average RE loss obtained during the training process. Accordingly, all the AE variants utilize their respective threshold value to detect the intrusion and their performance is compared with respect to the following comprehensive set of standard evaluation metrics (*Abdulhammed et al., 2018*).

a) *Accuracy (ACC):* measures the proportion of network traffic flows that are correctly classified and is computed as follows,

$$ACC = \frac{TP + TN}{TP + TN + FP + FN} \tag{11}$$

b) *Detection rate (DR):* Also called Recall or Sensitivity, measures the proportion of intrusion traffic flow that are correctly classified as given below,

$$DR = \frac{TP}{TP + FN} \tag{12}$$

c) *F1-measure (F1):* Also termed as F1-Score, is considered as more effective measure than accuracy to evaluate the performance of intrusion detection model especially for imbalanced datasets. It is an hormonic average of detection rate and precision as follows

$$F1 = \frac{2 \times (DR \times Precision)}{DR + Precision} \tag{13}$$

Here, precision measures the proportion of detected intrusion traffic that are actually correct. It is expressed as follows,

$$\text{Precision} = \frac{TP}{TP + FP} \tag{14}$$

d) *False alarm rate (FAR):* also termed as false positive rate, measures the proportion of normal network traffic flows that are incorrectly classified. It is computed as follows,

$$\text{FAR} = \frac{FP}{FP + TN} \tag{15}$$

### Implementation details

All the experiments are conducted on a personal computer with the specifications as follows, Intel Core i7-8565H @ 1.8 GHz, 128 GB RAM and Windows 10 operating system. The proposed model is implemented in Jupyter development environment using Python 3 as programing language. More specifically, the python libraries, Keras and Tensorflow are used to implement various deep learning tasks (*Géron, 2019*). Also, python Scikit-learn library is used to implement various evaluation measures and data preprocessing tasks.

## RESULTS AND DISCUSSION

This section compares and discusses the effectiveness of each of AE variant chosen in this study from three perspectives conducting experiments based on the framework illustrated in "Designed Experimental Framework for Comparison of AE Variants". Firstly, convergence ability of the chosen AE variants is carefully analyzed to signify the design decision of the compared AEs and to demonstrate their generalization ability on unseen attacks. As second step, the intrusion detection ability of the chosen AE variants is investigated to provide an insights into the applicability of different AE variants in practice. Finally, the performance of the chosen AE variants for imbalanced classification is examined to demonstrate the stability of the AE variants against imbalanced datasets.

### Comparison on network convergence ability

Procedurally, as a first step of experimental process, the training and testing sets of the both datasets discussed in "Datasets" are preprocessed according to steps described in "Data Preprocessing". Subsequently, for model training task, the processed training and testing sets of each dataset are fed into all the AE variants designed for comparison and 5-fold cross validation is conducted to assess their ability for converges and generalization on two datasets, NSL-KDD and UNSW-NB15 individually. This analysis is conducted from two point of view as follows.

#### Learning behavior

The graphical representation of training and validation loss against epochs for the last fold is displayed for all variants of AE for NSL-KDD dataset in Fig. 9.

Observing the training loss in these figures, it can be seen that all variants of AE show fast decrease in loss during first 3 epochs and also converges quickly while nearing epoch 7. Further, it can be observed from the curve of validation loss that all the examined AE variants offer better learning behavior to avoid overfitting and gain generalization ability. These observations evidently validates the optimal design configurations adopted for all

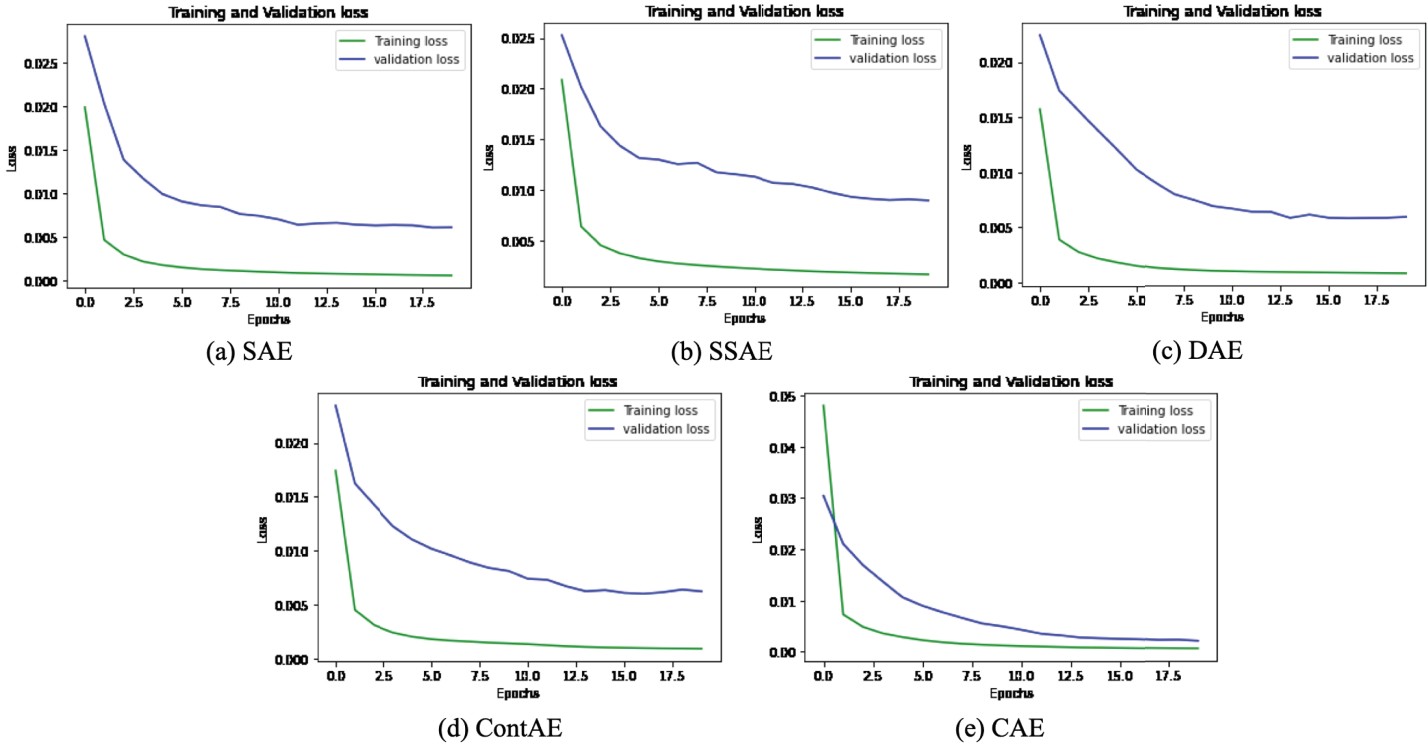

**Figure 9 Comparison of Loss curve on NSL-KDD Training dataset for different AE Variants (A) SAE (B) SSAE (C) DAE (D) ContAE and (E) CAE.**

**Table 6 Comparison of RE value for different AE variants.**

| Average RE | NSL-KDD | UNSW-NB15 |
|---|---|---|
| SAE | 0.0015 | 0.0022 |
| SSAE | 0.0012 | 0.0018 |
| DAE | 0.001 | 0.0038 |
| ContAE | 0.0003 | 0.0009 |
| CAE | 0.002 | 0.0022 |

AE variants and signify their detection capabilities for unseen attack styles in testing (validation) set.

### Reconstruction ability

To further quantitatively verify the reconstruction ability of each AE variant, the average of RE given by each AE during the training process is computed for both the datasets and reported in Table 6.

From these results, it is worth noticing that all the deep AE variants such as SAE, SSAE, DAE and ContAE demonstrates marginally better reconstruction ability compared to CAE variant. In many applications, convolutional networks have proved their outstanding performance, but it is surprise to note the average performance of CAE in this study. The reason may be due to the common network configuration that was followed across

**Table 7 Comparison of detection performance for different AE variants on NSL-KDD and UNSW-NB15 datasets.**

| AE Variants | NSL-KDD Training Set | | | NSL-KDD Testing Set | | | UNSW-NB15 Dataset | | |
|---|---|---|---|---|---|---|---|---|---|
| | ACC | DR | FAR | ACC | DR | FAR | ACC | DR | FAR |
| SAE | 89.61 | 97.26 | 17.04 | 85.23 | 85.13 | 14.62 | 87.16 | 96.58 | 32.90 |
| SSAE | 89.17 | 98.34 | 18.81 | 85.36 | 86.02 | 15.50 | 87.36 | 95.42 | 29.80 |
| DAE | 90.50 | 96.98 | 15.13 | 86.92 | 86.34 | 12.25 | 86.05 | 97.83 | 43.31 |
| lightgray ContAE | 91.46 | 97.00 | 13.34 | 87.98 | 89.23 | 13.66 | 88.48 | 96.83 | 29.30 |
| CAE | 90.90 | 96.42 | 13.90 | 81.07 | 80.37 | 14.21 | 86.66 | 96.26 | 33.79 |

all chosen AE for comparison, did not support CAE to discover the most discriminative representation to improve its reconstruction ability; despite this configuration demonstrated stable converges and generalization ability during training process.

Among all the examined deep AE variants, ContAE yields the best reconstruction ability on both datasets. This success of ContAE might be attributed to the incorporation of contractive penalty term in cost function which ensures to capture the more generalizable representations for reconstructions. Also, it can be noted that the denoising training process has enabled DAE to learn more robust representation for reconstruction and perform better than SAE and SSAE.

## Comparison on detection performance

This section aims to compare the intrusion detection performance of the examined AE variants. In this direction, the average RE reported in Table 6 is utilized as a threshold of respective AE variants, to discriminate the network traffic as either intrusion or not. Then, using these thresholds, the performance of all examined AE variants are analyzed first on training set of NSL-KDD and later on testing set of NSL-KDD and finally on UNSW-NB15 dataset. These results are presented and compared from two different views as follows,

### Quantitative analysis

In this step of analysis, the detection performance metrics such as ACC, DR and FAR are computed for all the examined AE variants and reported in Table 7. Also, for clarity the best results for each dataset are highlighted in boldface.

Inspection of the results remark that no single examined AE variants achieves best performance with regard to all the compared metrics. In particular, it can be noted that ContAE variant attains the best ACC and higher than average performance in terms of DR and FAR on three datasets. Also, we can see the inconsistent performance of CAE and DAE variants across the datasets. For instance, CAE variant shows comparably better performance on training set of NSL-KDD. But, surprisingly it delivers the least performance on testing dataset of NSL-KDD despite its behavior during training process on testing dataset was appealing. Likewise, DAE variant that delivers marginally better performance on NSL-KDD dataset, underperforms on UNSW-NB15 dataset. On contrary,

it is interesting to notice SAE and SSAE variants showing consistent average performance across all datasets.

Indeed, ContAE has demonstrated best performance across all three datasets in terms of ACC, yet, it is not apparent to confirm on the basis of ACC which AE variant is perfectly better. Therein, in the following sections, we explore the comparison of the examined AE variants on different metrics to verify the effects of these AE variants and to identify the best AE variant for building IDS with better detection ability.

### ROC curve analysis

To more intuitively compare the intrusion detection performance of the examined AE variants, the results in Table 7 are analyzed using Receiver Operating Characteristic (ROC) curve (*Ruisánchez, Jiménez-Carvelo & Callao, 2020*). This is carried out mainly because ROC curve is regarded as an important metric for visualizing detection performance of an IDS model. Besides, it enables to compare different IDS models offering relative importance for both DR and FAR metrics, that are remarked as essential performance metric for an idle IDS in practice. Accordingly, the average REs displayed during the training process by each AE variant were utilized as a threshold to generate the ROC curve of the respective AE. Figure 10 visualizes the ROC curves of all the examined AE variants on the two benchmark datasets, NSL-KDD and UNSW-NB15. An idle ROC curve approaches close to the upper-left corner and indicates the perfect classification performance. Against this background, it is visually clear that ROC curves of all the examined AE variants are close to the upper-left corner on both datasets. A careful observation of ROC curves on both datasets reveals an interesting finding that all ROC curves on NSL-KDD datasets are more towards upper-left corner than on UNSW-NB15. This finding indicates that the performance of the examined AE variants are sensitive to the proportion of normal samples in training set thereby confirming our initial discussion that the lack of sufficient normal traffic samples for training can affect the network performance. Intuitively, it obviously highlights that the detection performance of the examined AE variants can be further improved in real network settings, as adequate normal traffic samples for training will not be a constraint.

Subsequently, the area under the ROC curve (AUC) that quantitatively exposes the generalization ability of a model to recognize new attacks is computed and presented in the legend section of Fig. 10 for both datasets. As may be observed, the ContAE variant exhibits a reliable best performance on all three datasets with AUC value of (91.8, 87.8 and 83.8). On contrary, the variants DAE and CAE yield varying performance across three datasets with AUC values of (90.9, 87, 78.3) and (90.5, 81.6, 81.2) respectively. In general, the AUC results are in accordance to the results reported in Table 7 revealing the generalization potential of all the examined AE variants to gain better detection performance for abnormal network traffic samples.

### Comparison on imbalanced classification

In general, the intrusion datasets are imbalanced with less number of infrequent attack traffic samples and large number of normal traffic samples as illustrated in Tables 3 and 4.

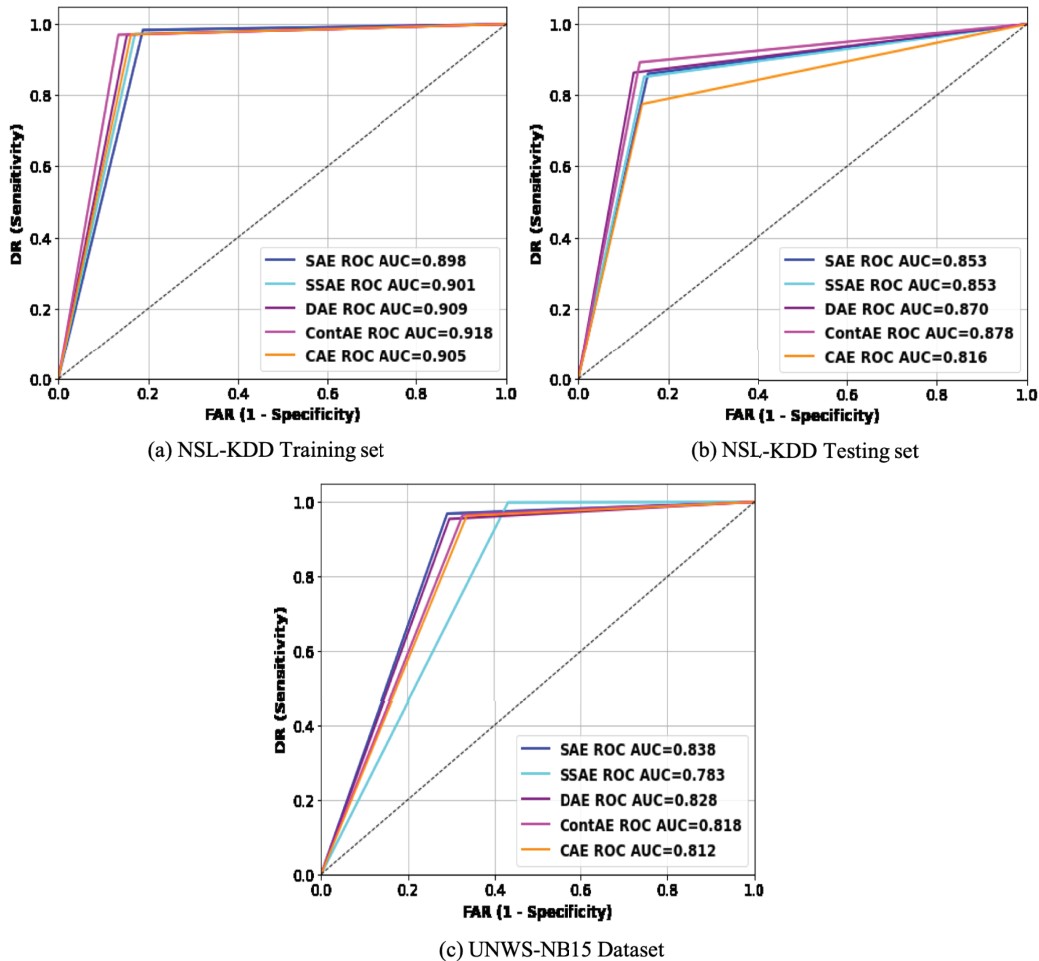

(a) NSL-KDD Training set      (b) NSL-KDD Testing set

(c) UNWS-NB15 Dataset

**Figure 10 Comparison of ROC curves for different AE variants on (A) NSL-KDD training set (B) NSL-KDD testing set and (C) UNSW-NB15 datasets.**

The presence of such circumstances intend the need to compare the behavior of the examined AE variants for imbalanced binary classification. Further, the standard metrics employed to analyze the detection performance in the previous section might not be appropriate to verify the imbalanced classification performance. For instance, ROC and AUC are valuable metrics for assessing the performance of intrusion detection model. But are insensitive to data imbalance and might tend to mislead giving optimistic performance for normal traffic samples. Furthermore, they fail to consider the model precision. Towards this end, this experimental analysis employs the most widely used metrics such as precision-recall (PR) curve, Area under PR (AUPR) curve and F1-score to provide a comparison of the examined AE variants with imbalanced intrusion datasets in the following sections.

### Precision recall curve analysis

Precision recall curve is a 2D graph that displays the relative trade-off between precision and DR. Recent literature recommend PR curve over ROC curve for two-fold reasons

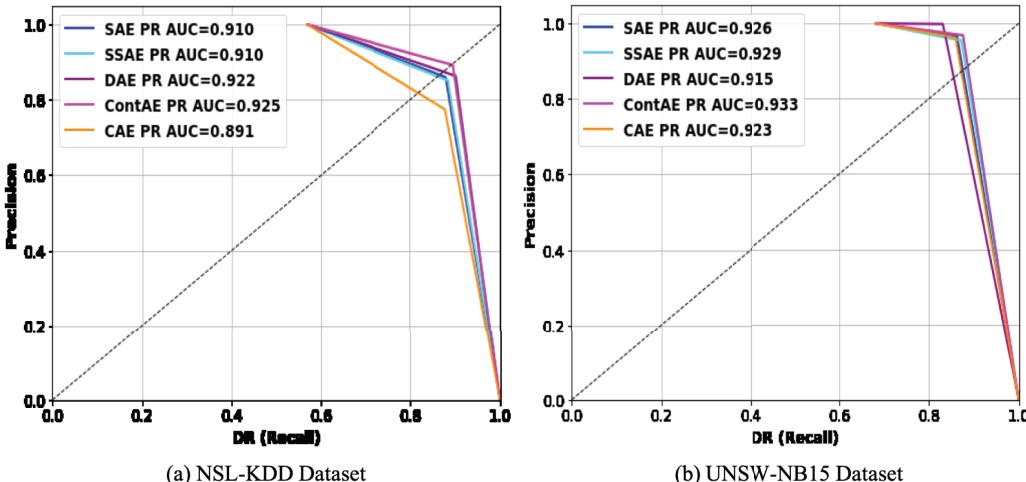

(a) NSL-KDD Dataset          (b) UNSW-NB15 Dataset

**Figure 11 Comparison of PR curves for different AE variants on (A) NSL-KDD and (B) UNSW-NB15 datasets.** 

(*Soleymani, Granger & Fumera, 2020*): First, it enables to measure the proportion of correctness to completeness when an imbalanced dataset is considered for classification. Second, it enables compare several classifiers and identify the classifier that maximizes intrusion detection precision with an acceptable DR. Realizing its advantages, the comparison of PR curves for examined AE variants on testing set of NSL-KDD and UNSW-NB15 dataset are illustrated in Fig. 11 respectively.

The PR curve for an IDS model with perfect discrimination passes through a point nearer to the upper-right corner. But for a model with no discriminability, the PR curve coincides with the diagonal line. Analyzing these figures from this perspective, it can be observed that PR curves of all the examined AE variants on both datasets passes close to upper-right corner. In specific, looking thoroughly the PR curves on NSL-KDD dataset, it can be found that all the examined AE variants are close to upper-right corner except CAE. Similarly, on UNSW-NB15 datasets, DAE shows exception with penalized performance compared to other variants. Further, it is worth noticing that the obtained PR curve results are also in agreement with ROC curve results on both datasets. From this observation, it is apparent that all the examined AE variants being trained only with normal traffic samples are capable of exhibiting reliable performance even against imbalanced intrusion datasets, thereby suggesting that the examined AE variants can serve as better choice as an one-class classifier to build an IDS with better detection ability.

To give more quantitative comparison, the AUPR values achieved by each of the examined AE variants on NSL-KDD and UNSW-NB15 datasets are computed and shown in the legend section of Fig. 11. The obtained AUPR values are also in agreement with AUC values indicating that all the examined AE variants are comparably efficient in achieving promising results in terms of precision and DR on both datasets. For instance, ContAE achieves the best AUPR values on both NSL-KDD and UNSW-NB15 datasets of (0.925, 0.933) in comparison to other examined AE variants. Also, it is interesting to observe that despite CAE delivers least AUPR value of 0.891 on NSL-KDD, displays above

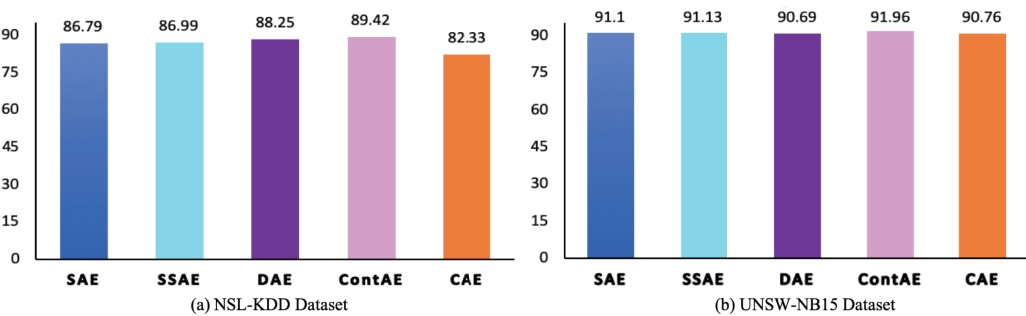

**Figure 12 Comparison of F1 measure for different AE variants on (A) NSL-KDD and (B) UNSW-NB15 datasets.**

average AUPR value of 0.923 on UNSW-NB15. On contrary, it can be seen that DAE displays better AUPR value on NSL-KDD but underperforms on UNSW-NB15 delivering AUPR value of 91.5. On the whole, the observation of PR curve and AUPR results indicates that all the examined AE variants are very efficient to gain better intrusion detection performance as one-class classifier against highly imbalanced datasets. This promising success in their performance might be attributed to the unsupervised training with only normal samples.

### F1-measure analysis

The F1 is an essential performance metric to highlight the efficiency of IDS model in terms of precision and recall using a factor that controls their relative significance (*Soleymani, Granger & Fumera, 2020*). Besides, in a most recent literature, it is pointed out that it can benefit over other metrics in favoring the correct classification of abnormal traffic samples at same time avoiding the misclassification of normal traffic samples. Accordingly, to gain deeper insight on the discriminative ability of the examined AE variants, the F1 is analyzed and the results are illustrated in Fig. 12. The obtained F1 again complements the above computed metrics and confirm the significance of the all examined AE variants in achieving better performance for intrusion detection. In particular, it can be noted that despite CAE displays low F1 of 82.33 compared to other AEs on NSL-KDD dataset, but still proves to display a better score of 90.76 on a highly imbalanced UNSW-NB15 dataset. In the same vein, other examined AE variants SAE, SSAE and ContAE except DAE also prove to gain better F1 on highly imbalanced UNSW-NB15 datasets. On contrary, DAE shows a F1 of 88.23 on NSL-KDD but a slightly penalized score of 90.69 on UNSW-NB15.

### Statistical analysis

As stated in literature (*Agbolade et al., 2020*; *Phillips et al., 2020*), we conducted statistical analysis to investigate the significant performance of different AE variants for intrusion detection. In this line, various statistics are computed on AUC, AUPR and F1-measures illustrated in Figs. 10–12 respectively for both NSL-KDD and UNSW-NB15 datasets. The computed statistics are presented in Table 8. Here, the best mean values are underlined for each metrics for clarity purpose. Observing these results in Table 8, it can be

**Table 8 Statistics of performance metrics (AUC, AUPR, F1) for different AE variants.**

| Evaluation metrics | AE variants | N | Mean | Std. Dev. | Std. error | 95% CI for Mean | |
|---|---|---|---|---|---|---|---|
| | | | | | | Lower bound | Upper bound |
| AUC | SAE | 2 | 83.55 | 2.47 | 0.78 | 82.01 | 85.08 |
| | SSAE | 2 | 84.05 | 1.76 | 0.55 | 82.95 | 85.14 |
| | DAE | 2 | 82.65 | 6.15 | 1.94 | 78.83 | 86.46 |
| | ContAE | 2 | 85.8 | 2.82 | 0.89 | 84.04 | 87.55 |
| | CAE | 2 | 81.4 | 0.28 | 0.08 | 81.22 | 81.57 |
| AUPR | SAE | 2 | 91.8 | 1.13 | 0.35 | 91.09 | 92.5 |
| | SSAE | 2 | 91.95 | 1.34 | 0.42 | 91.11 | 92.78 |
| | DAE | 2 | 91.85 | 0.49 | 0.15 | 91.54 | 92.15 |
| | ContAE | 2 | 92.9 | 0.56 | 0.17 | 92.54 | 93.25 |
| | CAE | 2 | 90.7 | 2.26 | 0.71 | 89.29 | 92.1 |
| F1 | SAE | 2 | 88.94 | 3.04 | 0.96 | 87.05 | 90.83 |
| | SSAE | 2 | 89.06 | 2.92 | 0.92 | 87.24 | 90.87 |
| | DAE | 2 | 89.47 | 1.72 | 0.54 | 88.40 | 90.53 |
| | ContAE | 2 | 90.79 | 1.65 | 0.52 | 89.76 | 91.81 |
| | CAE | 2 | 86.54 | 5.96 | 0.188 | 82.85 | 90.23 |

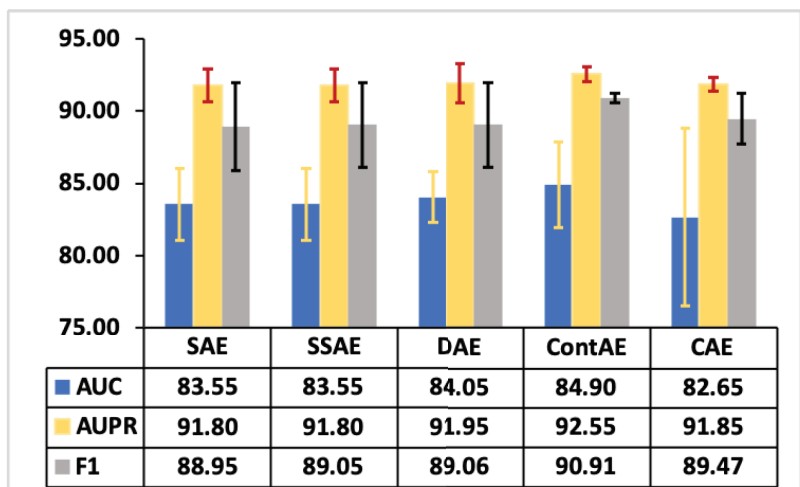

| | SAE | SSAE | DAE | ContAE | CAE |
|---|---|---|---|---|---|
| ■ AUC | 83.55 | 83.55 | 84.05 | 84.90 | 82.65 |
| AUPR | 91.80 | 91.80 | 91.95 | 92.55 | 91.85 |
| F1 | 88.95 | 89.05 | 89.06 | 90.91 | 89.47 |

**Figure 13 Mean plot of AUC, AUPR and F1 with error bars for different AE variants.**

noted that the minimum, maximum and 95% confidence interval (CI) for mean values of AUC, AUPR and F1-measure with regard to ContAE are higher compared to other variants.

Thus, it is evident that ContAE provides a better intrusion detection performance among all other variants. Nevertheless, looking at these means, it appear that a simple linear relationship did not exist. So, these mean values are plotted to provide a visual picture of the relationship with standard deviation as error bars in Fig. 13. Looking at this graph, one can note CAE displaying less error equivalent to ContAE with AUPR metric.

**Table 9 Comparison with related works on NSL-KDD datasets.**

| IDS Models | Training set | | | Testing set | | |
|---|---|---|---|---|---|---|
| | ACC | DR | F1 | ACC | DR | F1 |
| *Choi et al. (2019)* | 91.70[+] | 84.68 | 90.71 | NA | NA | NA |
| *Naseer et al. (2018)* | | NA | | 81.0 | NA[*] | NA |
| *Chen et al. (2020)* | | NA | | 85.02 | 86 | 80 |
| *Aygun & Yavuz (2017)* | | NA | | 88.28 | 87.86 | 89.51 |
| *Ieracitano et al. (2020)* | | NA | | 87 | 80.37 | 81.98 |
| *Shone et al. (2018)* | | NA | | 85.42 | 85.42 | 87.37 |
| Proposed | 91.46 | 97.00 | **91.36** | 87.98 | 89.23 | 89.62 |

**Note:**
[*] The corresponding metric is not available/provided in the published article.
[+] Highest score is highlighted with underline for each metrics.

However, its performance with regard to AUC and F1 are poor and discouraging. Also, it is interesting to see that a better intrusion detection performance is gained with ContAE in comparison to other AE variants with all metrics such as AUC, AUPR and F1. In this case, one can hypothesize that ContAE as one-class classifier has impacted the intrusion detection performance compared other AE variants.

## Comparison with related works

This section compares the results obtained utilizing AE as one class classifier IDS with some of the related works published in literature. To the best of our knowledge, very few works in literature have utilized the capabilities of AE for building unsupervised based IDS as we do in this study. Therefore, to examine the effectiveness of AE as one-classifier for IDS, the works related on AE based IDS are considered disregarding the variant of AE used and learning technique adopted for training the IDS. The results provided in their published articles are used to maintain fair comparison and this comparison results are presented in Table 9. Here, for clarity purpose, the highest score is highlighted in bold for each metrics.

Now observing the results, it can be realized that the utilizing AE as an one-classifier as in this study outperforms all the recent AE based IDS approaches for all metrics except for the two models introduced in *Choi et al. (2019)* and *Aygun & Yavuz (2017)* which displays the higher ACC value of 91.70 and 88.28 respectively. Though, these models show better ability in delivering high ACC value, its performance in terms of DR and F1 are very worst, essential metrics required for a IDS model. Hence, it is worth noticing that the use of AE as one-class classifier enables to gain better performance for intrusion detection than all other recent AE-based IDS approaches. Thereby we evidently recommend that AE as one-class classifier has great potential for building effective IDS with improved detection ability for unseen attacks.

## FINDINGS OF COMPARISON ANALYSIS

From the comparative results reported in the section, the following important insights are drawn

a) The optimal design decision of the common network architecture adopted across all variants of AE was verified for its significance and manifested.

b) Of the five AE variants compared for intrusion detection, ContAE yields the best detection performance in terms of ACC, AUC, AUPR and F1 across NSL-KDD and UNSW-NB15. This confirms that the incorporation of contractive penalty term into the cost function AE can enable capture the most robust feature representation for reconstruction and gain better performance for intrusion detection.

c) Convolutional AE exhibited inconsistent behavior across dataset with different characteristics. For instance,CAE delivered an average performance on UNSW-NB15 dataset but failed to provide consistent performance on NSL-KDD testing set despite it converges and generalization ability during training process was impressive on this dataset. The reason may be due to the common network architecture adopted for all AE variants was not optimal for CAE and thereby demanding more complex network model to capture useful feature representation from a dataset such as NSL-KDD which contains features with complex non-linear relationships.

d) Similarly, DAE demonstrated an average performance on NSL-KDD dataset but failed to provide reliable performance on UNSW-NB15 dataset. This difference in performance of DAE variant across the datasets may be due to the fact that for certain datasets even minimum noise level corrupts the useful information required for reconstruction; thereby affecting the detection performance of AE. Consequently, DAE is not a better choice of AE for such type of dataset.

e) Despite SAE and SSAE did not provide best performance compared to other chosen AE variants, these variants exhibited an average reliable performance across diverse datasets in terms of ACC, AUC, AUPR and F1.

f) Interestingly, all the AE variants examined in this study proved their detection ability for intrusion as an one-class classifier delivering higher DR and F1 value of 89.23 and 89.62 respectively on NSL-KDD dataset compared to the recent AE based IDS models reported in literature which ranged between 80% and 86% for DR and 81% and 89% for F1.

## CONCLUSION

A thorough comparison of five different AE variants for unsupervised IDS has been presented in this study. At first, the study provides an overview of all the AE variants selected for comparison which includes SAE, SSAE, DAE, ContAE and CAE. Second, to establish a common benchmark for fair comparison, all AE variants relied on unified network architecture and same datasets are used to evaluate the performance of different AE variants considering comprehensive set of evaluation metrics. The detailed comparative results demonstrate that all the AE variants offered comparable detection performance perfect to reconstruct the normal traffic samples and at the same time different AE variants displayed different RE value with regard to normal and intrusive network traffic which is also considered as threshold to discriminate the intrusion traffic

and evaluate their performance for intrusion detection. Also the effectiveness of each AE variant for imbalanced binary classification is investigated in terms of precision-recall curve, AUPR and F1 value. The summary of our analysis results is presented to provide an insight on the potential of different AE for intrusion detection. Hence, we expect this study to act as starting point to research further and improve the reconstruction ability of AE to build an unsupervised IDS that can gain even better detection performance for unseen attacks. Indeed our future work will focus to review different variants of variational autoencoders reported in recent literature (*Lopez-Martin et al., 2017*; *Lopez-Martin, Carro & Sanchez-Esguevillas, 2019*) and investigate their performance as one-class classifier for intrusion detection.

## ACKNOWLEDGEMENTS

The authors would like to thank the reviewers for their constructive comments and suggestions, which have improved the quality of this paper.

### Funding
This work was supported by the Research Deanship of Prince Sattam Bin Abdulaziz University, Al-kharj, Saudi Arabia. The funders had no role in study design, data collection and analysis, decision to publish, or preparation of the manuscript.

### Grant Disclosures
The following grant information was disclosed by the authors:
Prince Sattam Bin Abdulaziz University, Al-kharj, Saudi Arabia.

### Competing Interests
Thavavel Vaiyapuri and Adel Binbusayyis are IEEE members.

### Author Contributions
- Thavavel Vaiyapuri conceived and designed the experiments, performed the experiments, analyzed the data, performed the computation work, prepared figures and/or tables, authored or reviewed drafts of the paper, and approved the final draft.
- Adel Binbusayyis conceived and designed the experiments, performed the experiments, analyzed the data, performed the computation work, prepared figures and/or tables, authored or reviewed drafts of the paper, and approved the final draft.

### Data Availability
The raw code validated on public intrusion datasets is available as a Supplemental File.

## Supplemental Information

Supplemental information for this article can be found online at http://dx.doi.org/10.7717/peerj-cs.327#supplemental-information.

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
