# Peer review of "Application of deep autoencoder as an one-class classifier for unsupervised network intrusion detection: a comparative evaluation"

_PeerJ Computer Science, doi:10.7717/peerj-cs.327_

## Round 0.1 · original submission · Minor Revisions

The authors should address all comments of the reviewers and revise the manuscript accordingly. Specifically, the authors should discuss the reliability of their results, which should be supported by statistical analysis.

Reviewer 1 ·

Basic reporting

Interesting survey and comparison of autoencoder architectures used for one-class classification and their application to two well known IDS datasets. Some comments:
- The graphics are very clear and well done
- I understand that you opted for the deterministic version of autoencoders since variational autoencoders are not included, that is ok, but at least should be mentioned and included in the reference works. For example works such as: Variational data generative model for intrusion detection; Conditional variational autoencoder for prediction and feature recovery applied to intrusion detection in iot
- The quantitative analysis is well done and complete

Experimental design

The experimental design is relevant and complete providing convincing results

Validity of the findings

The study presents interesting results which are not themselves novel but the rigorous presentation and interesting survey can be useful to other researchers interested in the field of application of machine learning to IDS.

Additional comments

As mentioned earlier, it is an interesting work useful to the IDS research community. The only missing point is the inclusion of some additional reference to works in the field as pointed above

Reviewer 2 ·

Basic reporting

The article use clear, unambiguous, technically correct text. I don't feel qualified to judge about the English language and style. The list of references is appropriate.
Citations are not in acceptable format of ‘standard sections’. PeerJ uses the 'Name. Year' style with an alphabetized reference list, example (Smith et al., 2005).
Figures 10, 11 quality should be improved.
The paper ‘self-contained,’ represent an appropriate ‘unit of publication’, and include all results relevant to the hypothesis. The paper has no formal results.

Experimental design

The topic of the paper is appropriate for the journal PeerJ Computer Science.
Research question well defined, relevant & meaningful.
Methods described with sufficient information to be reproducible by another investigator.

Validity of the findings

It is clear from the paper what has been done and why. The scientific quality of the paper is good. The conclusions connected to the original question investigated, and are limited to those supported by the results.

Additional comments

Figure 2 what is “Avg”? If it means Average, explain why.
In equation 2 “N” is not defined.
In algorithm 1 Phase II: Testing, step 2 – “ifError” insert space after and clear Italic font – should be “if Error”, “else” must be bold.
In algorithm 1 Adam stochastic gradient optimization method is used. Authors should motivate why they use Adam method.
In case of Sparse autoencoder (SSAE) the sparsity penalty term is optimized minimizing the Kullback–Leibler (KL) divergence, but in algorithm 1 only Adam method mentioned.
In line 111 remove point, text in line 111“.Rifai et al. (2011)” should be “Rifai et al. (2011)”
For denoising autoencoder (DAE), Contractive Autoencoder (ContAE) and Convolutional Autoencoder optimization method used should be mentioned.
Tables 1 and 2 in columns “Value”, “Input size” and “Output size” data units should be shown, what is (41,1) percent’s, bytes or others?
Tables 1 and 2 described in section 3.5 but mentioned in section 4.1. The same is with tables 3 and 4, described in section 4.1 but mentioned in section 4.2.
The authors presents a comparative evaluation as mentioned in abstract “This study fills this gap of knowledge presenting a comparative evaluation of different AE variants for one-class unsupervised intrusion detection.” In addition, Experimental Framework proposed (see fig. 8). In such a case, it did not clear explained while and in which context sections “4.3.2 Model Training”, “4.3.3 Model Evaluation” of the paper became models? In the paper model did not proposed.

---

## Round 0.2 · accepted · Accept

Congratulations with accepted paper.